# Decision-making during obstetric emergencies: A narrative approach

**Gabriel M. Raoust**[1,2]*, **Johan Bergström**[3☯], **Maria Bolin**[4☯], **Stefan R. Hansson**[1,5]

**1** Department of Clinical Sciences Lund, Division of Obstetrics and Gynecology, Faculty of Medicine, Lund University, Lund, Sweden, **2** Women's Health Clinic, Ystad Hospital, Ystad, Sweden, **3** Division for Risk Management and Societal Safety, Faculty of Engineering, Lund University, Lund, Sweden, **4** Department of Applied Information Technology, University of Gothenburg, Gothenburg, Sweden, **5** Women's Health Clinic, Skåne University Hospital, Lund, Sweden

☯ These authors contributed equally to this work.

* gabriel.raoust@med.lu.se

**Data Availability Statement:** Data relevant to this study are available on the Swedish National Data Service at https://snd.gu.se/en/catalogue/study/2021-315 (https://doi.org/10.5878/ejjg-8477).

## Abstract

This study aims to explore how physicians make sense of and give meaning to their decision-making during obstetric emergencies. Childbirth is considered safe in the wealthiest parts of the world. However, variations in both intervention rates and delivery outcomes have been found between countries and between maternity units of the same country. Interventions can prevent neonatal and maternal morbidity but may cause avoidable harm if performed without medical indication. To gain insight into the possible causes of this variation, we turned to first-person perspectives, and particularly physicians' as they hold a central role in the obstetric team. This study was conducted at four maternity units in the southern region of Sweden. Using a narrative approach, individual in-depth interviews ignited by retelling an event and supported by art images, were performed between Oct. 2018 and Feb. 2020. In total 17 obstetricians and gynecologists participated. An inductive thematic narrative analysis was used for interpreting the data. Eight themes were constructed: (a) feeling lonely, (b) awareness of time, (c) sense of responsibility, (d) keeping calm, (e) work experience, (f) attending midwife, (g) mind-set and setting, and (h) hedging. Three decision-making perspectives were constructed: (I) individual-centered strategy, (II) dialogue-distributed process, and (III) chaotic flow-orientation. This study shows how various psychological and organizational conditions synergize with physicians during decision-making. It also indicates how physicians gave decision-making meaning through individual motivations and rationales, expressed as a perspective. Finally, the study also suggests that decision-making evolves with experience, and over time. The findings have significance for teamwork, team training, patient safety and for education of trainees.

## Introduction

Childbirth is generally considered to be safe in the wealthiest parts of the world but also show some puzzling variations between countries in both intervention rates and outcomes [1]. These international variations may be explained by differences in maternal and perinatal

**Funding:** - Initials of the author who received the award: SRH - Grant number: 20190142 - Full name of the funder: The Kamprad Family Foundation for Entrepreneurship, Research and Charity - URL: https://familjenkampradsstiftelse.se - The funders had no role in study design, data collection and analysis, decision to publish, or preparation of the manuscript.

**Competing interests:** The authors have declared that no competing interests exist.

characteristics as well as healthcare systems [2–5]. However, unexplained variations persist between different maternity units within high-income countries with standardized care, universal coverage and after adjusting for differences in the population [6–8]. It has been suggested that variations in outcome rates within a country may indicate an inappropriate use of interventions [3, 7, 9]. Indeed, as interventions may prove important in preventing neonatal and maternal morbidity [10], they may also cause avoidable harm if performed without medical indication [1, 11, 12]. Traditionally, the justification for an appropriate course of action in obstetric care is often grounded in an evidence-based approach [1, 10, 13]. The implementation of evidence is seen in best practice guidelines and checklists [14–16]. Furthermore, two other views originating in the decision-making and safety science research are slowly being incorporated into obstetric practice and physician training: a psychological view and/or a 'teamwork view' [17–19]. One view, borrowed from different schools of psychology, sees physicians as autonomous agents and focuses on the tools that can reduce errors and improve decision accuracy, or even practitioners' (i.e., obstetricians and gynecologists, and midwives) attitudes [20–23]. Training individuals in uncovering their biases (i.e., underlying affects and patterns of thinking) or 'risky attitudes' are implementations of such a psychological view of the problem [14, 20–22]. The other view puts more emphasis on teams and teamwork [24–28], and teams have been described as resilient, defined as Complex Adaptive Systems (CAS) [29]. Team performance is the result of interactions, relations and coordinative strategies among individuals connected to the team, rather than the result of "correct decisions" from a single person [27, 30]. This view is in support of team training and improved interpersonal communication [31, 32]. However, none of these approaches has yet been able to adequately resolve the existing problem of variation. As a new and complementary approach in trying to understand variation in interventions, we wanted to explore first-person perspective on decision-making; particularly those of physicians who hold a central role in the obstetric team as they are assumed to make key decisions before and during childbirth, especially during emergency situations. The purpose of this study is to explore how physicians make sense of and give meaning to their decision-making process during obstetric emergencies [33]. We believe that stories told by obstetricians and gynecologists to justify their decision-making process might enable us to interpret variation in terms of decision-making strategies and perspectives.

## Materials and methods

### Study design

A narrative approach using individual in-depth interviews ignited by the retelling of an event and having art images as associative stimulation, was chosen for this study. Narrative analysis provides a unique possibility to understand how individuals give meaning to the complexities of their experience through stories [34–38]. Using a *thematic* narrative analysis outlined by Riessman, and further influenced by Czarniawska's interpretation, the intent was to eventually render disparate stories into a coherent whole through a dialogical process [34, 39, 40].

### Ethical considerations

The study was approved by the regional ethics review board (Lund University, permit number LU 2018/198). The participants gave their written consent after being informed about the study and the voluntary nature of their participation. Participation could be terminated at any time. The interviewees were also assured of their anonymity and the strict confidentiality in handling interviews and results. Because retelling an obstetric emergency can bring back difficult memories, efforts were taken by the interviewer to create a compassionate and empathetic atmosphere during interviews [41, 42]. Considering some memories could be of traumatic

nature, collaboration with a psychotherapist was established during the study. No interviewee expressed a need for special counseling.

## Childbirth in Sweden

Intrapartum care in Sweden is institutionalized and publicly funded. There are 40 maternity units, and about 115.000 to 120.000 births per year serving a population of 10,2 million (2019) [43]. Pregnancy, labor, and childbirth are first and foremost considered natural processes. In 2020, 88% of births were so called normal births. Normal births are births without greater intervention: no Caesarean sections (CS) or no vacuum extraction (VE)/forceps, postpartum bleeding below 1500 mL or no need for blood transfusion, no sphincter rupture, no Apgar score below 7 at 5 minutes [43]. The care is standardized by national and local guidelines. Trained and autonomous midwives follow most of patients during pregnancy and assist during labor and childbirth. Midwifery is a university degree constructed around the medical sciences and modern medicine, and a previous degree in nursing is required. Physicians, trainees or specialists in obstetrics and gynecology are primarily involved in the care of patients with complicated pregnancies or when complications occur during labor and childbirth. Physicians take over the medical responsibility from the midwife when getting involved during pregnancy, labor, or childbirth. Historically, a non-interventionist ideal, i.e., wait and see rather than intervene, has guided practice for both midwives and obstetricians in Sweden [44]. The relationship between physicians and midwives in Sweden has also been characterized by considerably more teamwork rather than conflicts [44].

## Sampling and study population

The study was conducted at four maternity units in the southern region of Sweden. The maternity units used similar guidelines, routines, and practices. A combination of purposive and by referral sampling of obstetricians and gynecologists actively working with delivery care was used. Contact was established by e-mail. The selection of participants aimed at being representative of gender, work experience and preference for either obstetrics or gynecology. In total, 16 specialists and one trainee, of which 7 men and 10 women, were included. Another three physicians were asked to participate but declined. The most junior physician had worked 3 years and the most senior 31 years (mean age: 45 years ± 8.7 years, mean years of experience: 15 ± 9.4 years). Ten of the participants were working predominantly with obstetrics, 2 with gynecology and 5 had yet no preference. Inclusion stopped when no new data was generated by further interviews, defined as obtaining saturation [45, 46]. The interviewer and participants were known to each other on a professional level after having worked together to various degrees over the years.

## The interviewer

The interviewer, the first author (GMR), is board certified specialist of obstetrics and gynecology since 2009 and a senior consultant in perinatology since 2014.

## Data collection

The first author conducted all interviews, from Oct. 2018 to Feb. 2020. The time and location for the interview was agreed upon together with each interviewee (secluded places at the hospitals, research labs or in private homes). All interviews were carried out in a co-creative narrative form [41] (i.e., interviews were constructed and given meaning jointly by interviewer and interviewee), and divided in two parts. The first half was prompted by the question: "can you

recall an obstetric emergency situation that has left a memorable impression?" The second half
was initiated after the interviewees had glanced through randomly selected books depicting
images of various works of art. The interviewees were requested to find one or several images
speaking to them about decision-making during obstetric emergencies, more specifically: "try
to feel that the image is finding you, not that you are trying to fit your ideas into a particular
image." The interviewees were subsequently asked to draw a picture of the image, part of the
image or a synthesis of several images, using crayons. Conversation about the images and the
drawing ensued. Visual materials have shown to be effective in generating valuable informa-
tion that other methods fail to do [47, 48]. Among other advantages, using images and talking
about them can; I) level the field between interviewee and interviewer [49], II) "make the
familiar unfamiliar" again for the insider [50], III) disclose the unperceived in an experience-
taken-for-granted and IV) benefit the co-creative aspect of interviewing [51]. Here, images
and drawings were used to evoke new ways of thinking and talking freed from ideas and con-
cepts overly repeated within the profession [38, 48]. The interviews were generally unstruc-
tured and free flowing. The interviewer embraced a non-confronting attitude towards the
interviewee and actively listened to each one [42, 52]. Questions were asked during the inter-
view in order to clarify or to further develop the interviewees' ideas [42, 52]. Additionally, the
interviewer occasionally shared his own thoughts in order to further explore a train of
thoughts, however always empathizing with the participant [42, 52]. An interview guide was
only used in the rare cases of a stalled conversation [42, 52]. The interviews were audiotaped
and lasted from 35 to 97 min. (mean = 63 min.).

## Data analysis

All interviews were initially transcribed verbatim (including pauses, onomatopoeias, sighs,
laughs, and other sounds). The text was condensed and unorganized spoken language trans-
formed into a more readable form [34, 53]. Preservation of the internal coherence and meaning
of what was said was sought for [54]. The polished version of the transcripts were sent to each
interviewee to check for validity [34, 53]. An inductive approach of *thematic* narrative analysis
was used for interpreting the data [34, 53]. The interpretation already occurring as part of the
conversation, and now embedded in the transcript, was further systematized [34, 53]. By focus-
ing on "what" was said in the text, a search for patterns, differences and similarities within and
between the interviews was performed through a process of reading and re-reading [34, 53].
Intact narrative segments, defined as a bounded section of text about decision-making, were
identified [34, 53] and labeled according to what ideas were expressed. Ideas from across inter-
views were clustered into recurrent themes. Each transcript was also treated as an independent
whole, with the assumption that physicians had something meaningful to say about themselves
in how they made decisions. Ideas within each transcript were therefore also clustered indepen-
dently into an overarching meaningful perspective on decision-making. Furthermore, a tentative
interpretation of each interview was also presented to the respective participants for comments
[34, 53]. The process of interpretation as an oscillating movement, back and forth between the
whole and its parts [42] was also developed in collaboration with the other authors [34, 53].

## Results and analysis

The purpose of this study was to explore how physicians make sense of and give meaning to
their decision-making during obstetric emergencies. Through a process of narrative analysis
[40, 55] and by using the transcribed data as a whole, eight recurring themes across the inter-
views were constructed: (a) feeling lonely, (b) awareness of time, (c) sense of responsibility, (d)
keeping calm, (e) work experience, (f) the attending midwife, (g) mind-set (thoughts, moods

and expectations) and setting (time, place and circumstances) and (h) hedging. Additionally, three decision-making perspectives were constructed: (I) individual-centered strategy, (II) dialogue-distributed process, and (III) chaotic flow-orientation. The decision-making perspectives were the expression of physicians' identities transformed into practice, influenced by the different themes. The themes pertain to the environment common to all interviewees and are therefore presented before the decision-making perspectives.

## Themes

### a. Feeling lonely.

"*You are very lonely. When making a decision. You are often very, very lonely.*" (Physician 9)

The physicians felt lonely most of the time during obstetric emergencies. This was somewhat of a confession as current work-culture puts an almost compulsory emphasis on the team, teamwork, and cooperation. Nevertheless, feeling lonely was a reality with qualitative differences, ranging from factuality (being alone) to existential anxiety (loneliness) to empowering realization (aloneness). The physicians felt lonely with their decisions, the responsibility, and potential consequences from making the "wrong" decision. For some, the associated malaise of loneliness felt in early years would diminish with work experience. This was not a conscious choice but something that happened naturally over time. Maturation through personal, often distressing experiences helped in this process.

### b. Awareness of time.

"*The most important thing in the situation is to be aware of time and. . . Know that you have to, sometimes act quickly. Time, in some way, is always in the background.*" (Physician 14)

The awareness of time would most often become more acute as a situation became more pressing. Ultimately, time was a limiting factor that could put tremendous pressure on physicians with a demand for immediate action in the most extreme situations. As they gained work experience, they would develop their capacity for assessing the urgency of a situation and to modulate their responses. With lesser time pressure the physicians were also more likely to invite other team members to suggest ideas. In contrast, when time pressure was higher, the physicians felt a stronger need to regain control, and make decisions by themselves. A few of the physicians had the peculiarity of being able to move more fluidly between reclaiming control and openness, even to a point of letting go of control altogether, irrespective of the urgency.

### c. Sense of responsibility.

"*I have always felt a big responsibility, but. . . After I was involved in a situation in which the child got affected, I became much more aware of the responsibility. It became clearer somehow. . . And a little scary.*" (Physician 14)

The physicians were well aware of that childbirth is a natural process but considered it as a potentially high-risk situation. They all expressed a strong sense of responsibility, but the quality of the feeling could differ. Most of the physicians described responsibility as a burden, a source of stress accompanied by an underlying fear of not being able to manage. However, for some there was less interest in being the one that needed to solve "the problem". Medical responsibility was a given and focus had moved from fear to care, from concerns with oneself to concerns for the woman/couple giving birth, including for her/them to reclaim the experience. The physicians wanted their interventions to be remembered as less severe or exceptional than they were.

"*I think a lot about the couple. It's a very sensitive and important moment for them. It's great fun to be involved. . . But sometimes you get involved when things go awry. . . So you don't want to contribute to any terrible experience. . . I try to think a little long-term.*" (Physician 5)

Finally, some physicians felt the responsibility as empowering rather than crippling. By being engaged and integrated in the unfolding situation, they now had the capacity to respond. There was also awareness of how inter-dependent everyone involved were, seeing equal value in contribution from each one in the team.

"*When you've been in some emergency situations you become much humbler and understand the importance of cooperating. You learn about all the components of the process. You notice if there's a spanner in the works and you also know you were not alone when things go well.*" (Physician 17)

**d. Keeping calm.**

"*Back then I was only an observer. I couldn't help in any way and. . . I saw the reactions from the others in the room. . . How they became more and more stressed, after my colleague had said: 'I can't get the baby out, I can't get the baby out. . .' Especially after he lost it and started shouting: 'I can't get the baby out! I can't get the baby out!' And then I saw. . . You know the mother also heard.*" (Physician 13)

Keeping calm was a quality the physicians held in high regard and continued to want to develop throughout their career. As trainees, some had even experienced situations in which the colleague in charge lost their cool, leaving a traumatic impression as a numbing panic spread to other team members. For most, keeping calm was a purposeful commitment. It was about keeping the stressful thoughts and emotions in control or simply to pretend that every-thing was in order not to aggravate the situation. Projecting a sense of assurance or being per-ceived as cool enabled others to give to their fullest ability and for the woman/couple giving birth to feel safe enough regardless of circumstances. Others' apparent sense of calm and safety would give positive feedback to the physicians who would in turn start feeling calmer.

"*I try to radiate. . . That I control the situation even if it doesn't feel that way. It's important to try to convey that, so as not to create anxiety, especially for the parents. Anxiety is conta-gious. But if you're calm instead, that can also spread.*" (Physician 14)

A couple of the physicians expressed less conscious struggle with keeping calm. One physi-cian felt innately calm and had always done so. The other one expressed calm to be a by-prod-uct of being fully immersed in the moment.

**e. Work experience.**

"*In many situations. . . I just follow a routine. I've been exposed to the same situations so many times. And just act without further reflection. . . It works well. . . For sure. Then it's not about decisions. Well, yes. . . You still make a decision. But it comes so fast 'cause you just rec-ognize the situation.*" (Physician 16)

Having gained work experience, the physicians had invariably developed both routines and know-how. A resulting ease heightened the threshold for what was critical and enabled a greater range of maneuvering within each situation. The more experienced physicians also

expressed a better appreciation for nuances and subtleties. They were more interested in getting more information. Seemingly insignificant or even cumbersome bits of information for a beginner could be determinative for a more experienced person. Eventually some of the most experienced physicians had developed an intuitive capacity that they used in decision-making during obstetric emergencies.

"*Finding different patterns. What information you want and from whom and boil it down to a decision. But it's difficult to know what causes it sometimes, even though external factors are often quite similar, you do a little, little different. . . It eventually happens when you've worked for a long time, that sometimes you can't really say. . . Why you did this or that.*" (Physician 12)

Notably this was also more than know-how and pattern recognition. Intuition included the whole person projecting her conscious awareness (encompassing memories and experience) into the unfolding situation. Thus, participating in the construction of new possibilities for decision-making.

"*We fill the gaps with what we bring with us. With the knowledge and experience we have. And if you've worked for a long time, you can handle more bits of the puzzle.*" (Physician 12)

**f. The attending midwife.**

"*Midwives, it's part of it. . . Just like fetal heart monitoring or clinical examination and stuff. . . Sometimes when we're thrown into a situation, we might not even have time. . . To read the patient's record. . . So then you have to rely on what the midwife tells you in those twenty seconds. . . So I think it's an important. . . Part of our job. . . To have a good relationship with them.*" (Physician 10)

In Sweden, midwives are the primary caregivers during uncomplicated pregnancies, labor, and childbirth. It is only when complications arise that a physician gets involved, either through temporary consultation or by taking over the responsibility altogether. In an emergency this could result in a particular intervention such as,VE/forceps or a CS. Because midwives both have authority over the uncomplicated, sometimes also labeled normal, births and spend the most time with patients, the physicians felt awkwardly dependent on them for their decisions. Indeed, even if guidelines helped determine normality it was up to the midwife to involve the physician on call. Once the physician got involved, the midwife also decided how much information she wanted to share. The physicians had to negotiate their role and identity in relationship to midwives; first as an implied part of their training and then later as new staff got hired or if they changed workplace.

"*The relationship with the midwife is perhaps most problematic. . . And really, both if we think alike or and if we think differently. If we think the same, it affects me in my process because it makes it easier for me to make decisions. But if we think differently, it might make it harder. Because then I get a little worried that there will be a conflict.*" (Physician 2)

In practice, the physicians experienced midwives as gatekeepers, overtly or covertly preventing them from intervening because of a fear that it would disrupt the natural process of giving birth. At times, formal transfer of the medical responsibility could also be ambiguous, and physicians felt mislead instead of empowered. Ambiguity arose when physicians

experienced that it was unclear if the midwife wanted them to be involved or if she only followed guidelines, but actually did not want to involve them. Ambiguity also emerged when transfer was not about medical responsibility, but rather about emotions such as uncertainty and fear, in the guise of medical responsibility.

*"It's a special dynamic. . . You have to weight her reactions somehow, because she still has control over the patient. And been there. . . And everything. At the same time, you still have to be a little immune to that emotional game. So both and. Because I'm the one deciding. So I have to relate to the things she conveys that can be valuable information. It can be both factual and emotional, but also. . . Her opinions and will in this, and her thoughts on how the process should move forward. You can't let yourself be dragged along either. What you feel when you work with the midwife. . . If you have a lot of past history together. And if you have a collaboration with someone you don't really trust. It's a whole different story than if you work with someone you know. . . 'No! But. . . Her assessments usually make sense.'"* (Physician 9)

**g. Mind-set and setting.**

*"It can be how I feel that day. . . What mindset I bring with me or what I've done earlier. There are lots of things that can affect. How tired you are. The impressions you get in the situation."* (Physician 9)

The mind-set and setting were inner and outer circumstances affecting decision-making. The inner circumstances were about moods, levels of energy and stress amongst others. The outer circumstances could be workload, time of day, conflicts at the workplace or disorder in the delivery room, etc. An obstetric emergency was seldom a bounded event. The time-horizon extended both backwards and forward through memories and expectations, respectively. An earlier personal conflict that had generated anxiety lowered the capacity for bearing uncertainty, or the prospect of having to confront an interrogation from senior colleagues the next morning had led to a safer decision. The physicians coped with circumstances during decision-making in various ways. Many simply endured most of them. Treating every obstetric emergency as a specific problem to be solved was also helpful.

*"It's this woman lying there that I have to focus on, now. . . First, I have to identify where I need to make a decision. And temporarily shield off the rest."* (Physician 16)

Some of the physicians were more susceptible to be affected by the circumstances than others. This was based on their sensitivity alone. In general, increasing the awareness around their sensitivity through self-reflection and conversations with colleagues and midwives outside of an emergency was usually helpful enough for coping. A few of the physicians had come to accept circumstances as a range of uncontrollable parameters integral to any emergency situation.

**h. Hedging.**

*"Much of the decision-making process is about being one step ahead. It's about anticipating or preparing, sometimes only in thought, for potential problems."* (Physician 6)

Hedging was about having alternatives and imagining possible scenarios on how a situation could develop. This was a mental event, mainly based on the guidelines' algorithms and/or knowledge from past experiences.

"*I had my inner checklist. And it made me feel pretty safe. I felt I was little step ahead in my mind and. . . Felt that even though I hadn't done this many times, I had thought about it.*" (Physician 7)

But hedging was also about empowering the team through "thinking out loud" (i.e., the overtly sharing of thoughts). Beyond simply verbalizing the guidelines' algorithms it was an invitation for other team members to participate and share their perspectives and thoughts.

"*It's not that everyone is silent and waits for me to say something but rather like. . . Everyone helps with what they can bring to the table.*" (Physician 17)

Consequently, relationships between team members and trust within the team got strengthened. The patient/couple could sometimes be included in the discussion, even in the midst of an emergency. This was mainly for informative and pedagogical purposes but could also be as a preventive/therapeutic measure. Informing a patient of an upcoming intervention such as a VE could for example motivate her to push and give birth, consequently avoiding the intervention altogether. Finally hedging could be more of an attitude integrated in decision-making itself, less as different scenarios and alternatives and more as leaning into the next moment.

"*What's the best thing we can do based on the prevailing conditions. . . And in the next moment, and the moment after that. And then you have to be at it again, all the time: 'what's advisable now? How do I relate to that?'*" (Physician 3)

## Perspectives

### I. Individual-centered strategy.

"*I'm the one in charge. . . I'm the one that needs to make that decision.*" (Physician 13)

The individual-centered strategy as a decision-making perspective was preferred among 7 physicians. From this perspective, physicians took for granted that they were the focal point. All information, mostly external–visual or verbal–was perceived as input and processed in the black box of their mind eventually resulting in directives to others, who were mostly seen as instrumental. Teams and teamwork were perceived as ideological concepts rather than a way of working that reflected reality.

"*There's a lot of input and some mess. . . And then you want to try and get a little bit of order and structure and. . . Try to form it to something manageable. There's a process in the brain.*" (Physician 11)

Physicians having this perspective would rather function according to rules, and had a strong sense of right and wrong as well as a particular respect for authority. For them there was a good decision and a bad decision, and the closer to guidelines the better.

"*So, according to the guidelines I did right. And nobody can say I didn't.*" (Physician 15)

This was particularly significant when they had to deal with unfamiliar situations or make a decision when there was great uncertainty regarding the resulting outcome. For physicians with this perspective the best direction would always involve as little unknowns as possible, thus

minimizing the risk of a detrimental outcome and/or personal discomfort. Physicians felt they needed to regain control, even if it was immediately detrimental for the woman in labor. Performing a CS, which is a routine procedure in a controlled environment, rather than attempting a breech delivery, a vaginal birth considered to be a higher risk, was for example preferred.

> "*If everything goes well, that's good. But if it doesn't, I want to feel that I have that much to back it up... That it was the right decision. And, if something goes wrong... It should have been the right decision anyway.*" (Physician 11)

In summary, a profound sense of personal responsibility, loneliness, and a need to be in control were essential to physicians having this type of decision-making perspective. The parturient was often described as someone suffering and/or a helpless person in need of assistance. Physicians felt they got involved to help a victim in distress and that their actions would restore order from chaos.

> "*Patients are like angels. They can't do anything by themselves. Rather, it's us taking care of it. I mean the patients. Innocent... Right!? They are victim of the situation.*" (Physician 13)

**II. Dialogue-distributed process.**

> "*In earlier years I thought I needed to know everything, what to do, and what would happen. Nowadays, we make decisions together. We have a conversation. You don't need to be the one with all the answers and have a solution to every problem.*" (Physician 2)

The dialogue-distributed process as a decision-making perspective was preferred among 7 of the physicians. From this perspective physicians saw themselves as part of a team and were looking for dialogue with others involved, sometimes even including the patient and at times the partner/relative present in the room. In lesser emergencies, a senior colleague's advice could also be sought for. Furthermore, communication, mostly verbal but sometimes also through gazes and/or even body language was central in getting information or making headway with the situation.

> "*You communicate with touch and how you move. The body gives out signals. And you know what to do. It's rehearsed. It's not only verbal or visual. You can do your work with physical contact, that sort of communication... And there's an energy in working together that carries things forward. There's communion. And everybody's fully concentrated on what to do, but at the same time there's this sensitivity and wakefulness to each other.*" (Physician 2)

Generally speaking, decisions would rather be distributed or delegated. A commonly used tool was again "thinking out loud". Other team members would suggest actions and measures to take, and the physician would only need to give an approval, rather than solving the situation himself. Knowing each other on a personal level or simply having worked together previously was facilitating this type of decision-making. Notably, when physicians who used this type of communicative perspective were in a tense or conflict-laden situation they would still prefer to discuss, in order to solve the issue at hand, rather than reclaiming their medical authority or a need for control. In summary, the main characteristic of this perspective was a shift of attention from individual concerns to group-work, including the patient. And because of the feeling of meaning created in this communitarian micro-cosmos, engagement as a team was also stress reducing and at times even enjoyable.

*"It's also about getting relief from stress. I don't need to be busy with the thought of having to control everything, while having to make all the decisions. . . I know I can trust that others are doing what they should. It's like choreography."* (Physician 2)

**III. Chaotic flow-orientation.** The chaotic flow-orientation as a decision-making perspective was preferred among 3 of the physicians. What particularly characterized this perspective was dynamism and letting go. For these physicians this type of decision-making had emerged over time, through maturation, self-reflection, and a more personal development and not just from sheer work experience. Physicians who preferred this perspective also felt that they could modulate their responses and revert back to either one of the two perspectives previously described. They also expressed an understanding that fundamentally, obstetric emergencies are full of uncertainty and accepted it to be so.

*"We don't know when we make decisions. We don't have all the information. When everything is over, then you know. But, when you stand there, then that's just the way it is! Most of it is still unknown."* (Physician 12)

In combination with that, physicians also expressed an appreciation for their finitude as human beings, on the limitations of how much one can possibly know and do.

*"It's not always we have firm ground to stand on. Sometimes we're just treading water, even if we think we know what we're doing. We don't really have control over things."* (Physician 12)

From this perspective physicians' focus and perceived role in emergency situations had changed. Focus was no longer on oneself, the team, or the birthing woman anymore, as in the two other perspectives. Focus was rather on "something greater than oneself". The physicians' role was to make oneself and one's skills and knowledge available in an unfolding situation.

*"Believing in yourself when you are in front of the unsolvable is grounded in a trust that there's something higher, that you can't explain. You go into it with humility at the same time. You feel that you have a guaranty that it's going to be all right in the end. It's not dependent on you that much, it's rather: 'so be it!'"* (Physician 17)

Concomitantly, emergency situations were perceived as inclusive (i.e., physicians felt immersed in the experience, not standing apart from it). Physicians were part of both the solution and the problem. Emergency situations were also perceived as dynamic and participatory, (i.e., the physicians expressed an acceptance over the ever-changing nature of situations and that every participation from any team member mattered). No one could avoid participating.

*"I have to be there and see and say what I think. Because then others can also say what they think. And then it's easier to find the direction you need to take right there and then."* (Physician 3)

There was also an active and particular attention to detail, as if everything mattered, because solutions could come when most unexpected.

*"It's like a lightning bolt, suddenly you know what you must try. You get an insight: 'this I have to try! It moves things forward.'"* (Physician 17)

Ultimately, this perspective was grounded in a synergy between trust in the unfolding situation, as process and a responsibility for participating in that process. Physicians preferring this perspective had a belief that things would develop "as they should", with an understatement that the outcome would be favorable. At the same time, they knew fully well that it wouldn't necessarily be the case. Here physicians did not feel a need for control as in the first perspective. Instead, physicians expressed an appreciation for creativity and improvisation. Both were considered as necessary assets and what felt as a natural consequence of this decision-making perspective.

"*It's a creative process that's helping you, getting to an unconventional solution. You go through, unhinged, all possible alternatives. Your fantasy runs wild. There's no inhibition, no barrier.*" (Physician 17)

## Discussion

In maternity care, there is an assumption that everyone will act according to the same rules and norms in each situation [17–19, 32]. A plausible explanation for this comes from a historical perspective. In 1979 Archie Cochrane anecdotally awarded obstetrics the wooden spoon award for being the poorest of all medical specialties in evaluating its practices. As a result, the interest in evidence-based medicine and standardization of effective practice has grown stronger and stronger ever since [13]. Many of today's better outcomes of pregnancy and childbirth, for both woman and child, are often attributed to this development [1, 13, 15, 16]. Simultaneously, evidence that guidelines significantly improve health outcomes in obstetrics and gynecology is relatively scarce [56]. Researchers have also expressed reservation regarding the value of best practice for healthcare situations that show a great deal of variability, uncertainty and risk [57], such as emergency obstetrics. Our results of reconstructed narratives, expressed through themes and perspectives, indicate that physicians were, to a great extent, driven by individual motivations and rationales during their decision-making. This goes somewhat against the current trend of standardization. But most importantly, those motivations and rationales seemed to exist unnoticed, unless brought to the interviewee's awareness, with the help of retelling an event and discussing images. This raises a couple of questions. How do these unconscious themes and perspectives impact the dynamics of teamwork, the functioning of a ward or a clinic, and per extension influence outcomes during childbirth? Would there be any value or even tangible effects from making themes and perspectives more conscious?

### Recurrent themes as emergent discretionary space

The eight recurring themes were the most common elements physicians directly or indirectly related to during obstetric emergencies. Themes show how physicians both are continually constructed within-, and continuously constructing, a discretionary space for their decision-making [58, 59]. Constraints and possibilities are not simply determined by the preset limit-conditions of work but are also emergent properties of the physicians' own (inter)actions, relations and involvement with their immediate surrounding [60, 61]. To keep one's calm was volitionally self-imposed and yet often understood as a necessity, for different reasons such as keeping one's thoughts in check or not distressing the woman giving birth. The relationship with 'the attending midwife' both existed as a negotiable power-dynamic and as a factual condition. In other words, interactions between midwives and physicians were part of the way work was organized and how that interaction was enacted depended on both the midwife and the physician. A similar discussion regarding the balance between structure and agency aspects in each of the other themes could be developed. In general, our findings confirm previous

research showing that decision-making in critical situations is more complex than adherence to routines, regulations and procedures [27, 62]. Indeed, similar intricacies of work processes in the delivery room have been highlighted and problematized by other ethnographic studies pointing towards often unexamined and more complex dimensions of decision-making in obstetrics [63–65]. As shown in this study, environmental conditions–including human inter-actions, norms, and cultures–and the physicians' motivation and rationale engaged in a co-cre-ative process with one another [66, 67]. Nevertheless, in the quest for quality, safety and accountability in healthcare, the tendency has been to split reality of practice into agent and environment, practitioner and working conditions [17, 68]. We suggest that organizational attempts taking into account the deep entanglement of the two [68–70] are worthwhile exploring.

## Decision-making perspectives, identity creation and variation

The three preferred decision-making perspectives expressed by physicians were contingent upon how different themes weighted and were valued in any given situation. More precisely, perspectives were the unification of the physicians' disparate stories of professional identity and practice. Paradoxically, these stories of identity were often undisclosed to the self [71, 72]. The underlying perspectives appeared to be of importance in defining the physicians' actions in emergency situations in general and more specifically towards the other members of the team, and the woman/couple giving birthing. Several aspects of the three different decision-making perspectives have also been the target of previous research. Our findings, characterized as 'individual-centered strategy' parallel those of an agent centered research on cognitive and affective aspects of decision-making [20–23], those of 'dialogue-distributed process' parallel research on teams and teamwork [24, 26, 30, 73, 74], and those of 'chaotic flow-orientation' parallel research on effortless attention [75] and intuition [76–78]. As a whole, the findings resemble three of four decision-making styles described in a 1995 study by Scott and Bruce [79]. Four decision-making styles were identified through the literature and further tested into an assessment tool. The four styles were: (a) a rational decision-making style characterized by a thorough search for and logical evaluation of alternatives, (b) an intuitive decision-making style characterized by reliance and hunches and feelings, (c) a dependent decision-making style characterized by a search for advice and direction from others and (d) an avoidant deci-sion-making style characterized by attempts to avoid decision making [79]. Moreover, seen as a unified model the three perspectives are also akin to Dreyfus and Dreyfus' model of skill acquisition [80, 81] describing how professionals develop from novice to expert through differ-ent stages, how each stage gives rise to a certain understanding of the world and how individu-als express their stage specific skills. Indeed, interviewees hinted at a developmental aspect of decision-making suggesting that their perspective evolved from 'individual' to 'dialogue' to 'chaos' along with their professional and personal experience. Putting decision-making, as strategy, style, or perspective at the level of individuals into a structural framework of the development of expertise through experience would have implications, at least for trainee edu-cation. In fact, this has been highlighted in nursing research [81]. For example, novices show-ing a strict technical application of knowledge do not have the skills required for discerning the nature of a situation and its possibilities and constraints. Experts on the other hand, remain open to experiential learning and read changes in transitions in fast-paced, open-ended envi-ronments [81]. And most importantly, experts have shown to be able to act under time pres-sure, either after their quick recognition of a problem or by applying a strategy that has proven successful in similar situations [82]. Another theory worth mentioning here is Hammond's cognitive continuum theory that models clinical decision making on a spectrum between

intuitive and analytic modes [83–86]. Location on the spectrum is dependent on various factors such as type of task at hand (action versus planning), time available (short amount versus greater amount), type of knowledge available (unstructured versus structured), and the kind of health problem needing to be dealt with (acute/unstable versus long-term/stable). The cognitive continuum theory parallels some of the physicians' narratives about their capacity to modulate their responses during emergencies (i.e., the more urgent the demand for an action was, the more intuitive their decision would be). Finally, even though three perspectives were characterized, most physicians rarely embraced one style exclusively. Individuals embodied, to a varying degree, a plurality of perspectives. Eventually, one perspective would dominate over the others, seemingly depending on personal characteristics and situational circumstances as exemplified through the themes. In terms of personal characteristics, physicians were more or less sensitive, and some possibly resistant to the impact of different themes. Examples from interviews, such as the statement of "being born naturally calm" or that some interviewees experienced difficulty in relating to images, suggest that individual-specific, non-contingent factors are at play during their decision-making. It has been suggested that personality, the unchangeable characteristics of an individual, could be such a factor [21, 87, 88]. In fact, studies have shown that individual propensity for anxiety, type of coping and adaptive traits are associated with perinatal outcomes [21, 87, 88]. Furthermore, it has also been suggested that other traits facilitating collaboration, coordination, cooperation, and participation impact team effectiveness, team member satisfaction and achievement of better patient outcomes [73]. There is a considerable amount of research on the subject of individual differences (e.g., risk behavior tendency, sensitivity to situational factors, personality, etc.) in relation to decision-making [89]. However, this work has generally been confined to psychology and more specifically to the research field of judgment and decision-making. With a focus on patient safety and optimizing care, the interest in decision-making applied to healthcare has rather been on trying to minimize the impact of individual differences and, to some extent, understand the impact of situational factors. Further research on the relationship between decision-making perspectives, themes as exemplified in this study, personality and the impact of individual traits in obstetric care is warranted.

## Methodological considerations

**Self-reflexivity.** The fact that he first author (GMR) was an experienced obstetrician provided certain advantages for the interviews. Acknowledging each other's expertise within the domain of interest established an atmosphere of trust during the interview situation. Shared contextual understanding was reinforced by the first author further acknowledging participants' responses, often from having had similar experiences to the ones described and having reflected upon them over the years. This contributed to an ease and flow during the interviews. Participants were enabled to open up more. Knowing the jargon provided a unique opportunity for reaching deeper faster with each interview and by-passing polite, superficial or expectable answers [90, 91]. Being an experienced obstetrician and being inherently embedded within the research context also conferred some difficulties both during the interview and analysis. Indeed, one will always be unable to fully appreciate one's own assumptions within the domain of interest. In this regard, our understanding and use of both the insider researcher approach and the narrative approach in this paper is grounded in the views of hermeneutic phenomenology [92, 93]. Accordingly, our preconceptions (i.e., knowledge, insights, and experience) of the world come from being inextricably involved within it and "stepping outside" or leaving our biases aside is understood as impossible [92, 94, 95]. However, through a dialogical process of interpretation, preconceptions and the creation of new meanings and an

understanding of the world can simultaneously be disclosed [95]. The general approach of the inquiry was co-creative [41]. The emerging data was constructed through unfolding conversations-*cum*-interpretation [53], with just enough involvement in facilitating the interviewees' responses [42]. Neither the interviewee nor the interviewer were neutral observers of a phenomenon to be ultimately revealed [90, 91], and no one was considered to have the true answers. It was rather two colleagues reflecting on each other's way of handling an emergency. In this context, meaning and reality were constructed through language [35, 53, 96, 97].

**On the use of visual materials.**   Overall, the use of images helped most interviewees in developing their responses, sometimes even generating new ideas. Yet, a few interviewees had difficulties choosing and talking about images, suggesting that this method is either not suitable for all or that special guidance or sensitivity on the part of the interviewer might be necessary. Notably, most physicians felt the drawing part to be initially distressing. However, as curiosity for the method and a desire to share their stories quickly evolved, they became more at ease and playful with the concept. The combination of retelling an event and using images was thought to provide particular richness to the interviews [98]. Eventually, analysis was derived from a bulk of data consisting of text.

**Truth and validity.**   The narrative approach is interpretive in its nature. We acknowledge that our original material is context-dependent and can be analyzed in different ways [99]. Furthermore, the results and analysis are influenced by the authors' respective background, and the first author's deep involvement in the process, as described above. Nevertheless, several methodological steps were taken in an attempt to create a trustworthy and valid reconstructed narrative in accordance with the participants responses [47]. Firstly, a narrative approach will have validity in its contribution to making sense of the world and ourselves [53]. Here, we subscribe to a *pragmatist* view [55, 100]. We believe the study has value for both clinical practitioners and researchers alike. Whether this research ultimately contributed anything of value is for the readers to judge [34, 40]. Secondly, coherence of interpretation for each interview and between interviews was sought for [54, 101]. We believe that both themes and perspectives form coherent wholes that make sense to practitioners. Thirdly, sending the interview and the written interpretation back to respective interviewee insured further validity [34, 36, 42]. Interpretation was found to be in accordance with participants' answers. Fourthly, the use of visual materials functioned as a neutral ground for inquiry, effective in generating valuable information to some degree freed from preconceptions [38, 47–51]. Finally, having an ongoing discussion about different findings amongst the four co-authors was a way to keep the interpretation alive [53, 102]. The authors' different backgrounds and perspectives fueled new reflections.

## Conclusions

The purpose of this study was to explore physicians' sense-making of their decision-making during obstetric emergencies. A co-creative narrative approach was used. The analysis interpreted themes of how physicians were both situated within-, and creators of, conditions in an ongoing dynamically adaptive process. Furthermore, the analysis indicated how physicians gave meaning to their decision-making through individual motivations and rationales, interpreted in the form of three perspectives: an individual-centered strategy, a dialogue-distributed process, and a chaotic flow-orientation. Finally, the analysis also suggested that an individual's perspective evolve with experience, and over time. Physicians' narratives confirm previous findings about the multilayered complexity of decision-making in intrapartum care. Various aspects of the reconstructed perspectives are reminiscent of previous findings on individual and team decision-making in emergency medical settings. However, we believe that

perspectives understood as identity, during decision-making in obstetric emergies, has previously been insufficiently explored. Overall, the findings have relevance for continuous efforts to improve clinical practice and contribute to our understanding of the variability of strategies and decision-making processes in intrapartum care. The narrative approach used, including visual materials, proved rewarding and might be well suited for similar research on social complexity in other healthcare areas.

## Supporting information

**S1 Appendix. Interview guide.** Interview guide used in the rare cases of a stalled conversation.
(PDF)

**S2 Appendix. Interview guide in Swedish.** Intervjuguide, används i de fallen samtalet stannar upp.
(DOCX)

**S3 Appendix. List of visual materials.** List of visual materials chosen by respondents.
(DOCX)

## Acknowledgments

The authors would like to thank the 17 obstetricians and gynecologists for their time, commitment, and interest in the study. The authors would also like to thank Dr. Karin Sjöström for providing psychotherapeutic counseling support during the study. The authors would also like to thank Dr. Lena Erlandsson for her thoughtful suggestions to the final version of the manuscript.

## Author Contributions

**Conceptualization:** Gabriel M. Raoust, Johan Bergström.

**Data curation:** Gabriel M. Raoust, Johan Bergström, Maria Bolin.

**Formal analysis:** Gabriel M. Raoust, Johan Bergström, Maria Bolin, Stefan R. Hansson.

**Funding acquisition:** Gabriel M. Raoust, Stefan R. Hansson.

**Investigation:** Gabriel M. Raoust, Johan Bergström, Maria Bolin.

**Methodology:** Gabriel M. Raoust, Johan Bergström, Maria Bolin.

**Project administration:** Gabriel M. Raoust, Stefan R. Hansson.

**Resources:** Gabriel M. Raoust, Stefan R. Hansson.

**Software:** Gabriel M. Raoust.

**Supervision:** Gabriel M. Raoust, Johan Bergström, Maria Bolin, Stefan R. Hansson.

**Validation:** Gabriel M. Raoust, Johan Bergström, Maria Bolin, Stefan R. Hansson.

**Visualization:** Gabriel M. Raoust.

**Writing – original draft:** Gabriel M. Raoust, Johan Bergström, Maria Bolin.

**Writing – review & editing:** Gabriel M. Raoust, Johan Bergström, Maria Bolin, Stefan R. Hansson.

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
