## [Decision Letter · Decision Letter 0]

28 Jun 2021

PONE-D-21-12199

Decision-making during obstetric emergencies: A narrative approach

PLOS ONE

Dear Dr. Raoust,

Thank you for submitting your manuscript to PLOS ONE. After careful consideration, we feel that it has merit but does not fully meet PLOS ONE’s publication criteria as it currently stands. Therefore, we invite you to submit a revised version of the manuscript that addresses the points raised during the review process.

We look forward to receiving your revised manuscript.

Kind regards,

Bernadette Watson, Ph.D.

Academic Editor

PLOS ONE

Journal Requirements:

3. Please include a copy of the interview guide used in the study, in both the original language and English, as Supporting Information, or include a citation if it has been published previously.

Additional Editor Comments (if provided):

I have now received feedback from the two reviewers. They both see the value in the paper but make very important points. While one reviewer suggest minor revisions, there is an important question posed about making sure your research question is clear. I think both reviewers' comments merit a major revise and resubmit, which I hope you will do. Please address their comments carefully. I look forward to your revisions.

Reviewers' comments:

Reviewer's Responses to Questions

**Comments to the Author**

1. Is the manuscript technically sound, and do the data support the conclusions?

Reviewer #1: Yes

Reviewer #2: No

2. Has the statistical analysis been performed appropriately and rigorously? 

Reviewer #1: N/A

Reviewer #2: N/A

3. Have the authors made all data underlying the findings in their manuscript fully available?

Reviewer #1: No

Reviewer #2: Yes

4. Is the manuscript presented in an intelligible fashion and written in standard English?

Reviewer #1: Yes

Reviewer #2: No

5. Review Comments to the Author

Reviewer #1: Thank you for the opportunity to review the paper "decision-making during obstetric emergencies: A narrative approach". Overall, I thought the paper was interesting and the methodology in particular was interesting and conducted with rigour. I do however feel that further work is required before the manuscript is published.

In particular, I feel that more needs to be done to justify why the study was needed. Why is there a need to understand how physicians make sense of and give meaning to their decision-making? And how will will understanding their sense making and meaning making processes help us to address a problem/issue. The problem/issue with our current understanding of decision making in emergencies was alluded to but not well defined. It seems like a gap that was trying to be filled was considering team-based decision making (which has evidently been used in anaesthesiology), but if that was the case, why were only Physician’s included? Given the plethora of literature on medical decision making, I felt that the introduction was too short and lacking critical analysis of the existing evidence base and thus lacking the justification for the current study.

Typically, qualitative research is used to answer a research question. I wasn’t clear what the research question was that was being answered in the study.

I thought the method and description of the approach was good, though I wasn't clear why median years experience was chosen rather than the traditional mean and standard deviation, which gives more indication of spread, was chosen. Age is also commonly reported though is perhaps less relevant for some reason in this study.

Given the study was a 3 different units, I was interested in whether the unit they worked in influenced the themes and style used. Perhaps this was considered and was not of interest as experience drove most of the effect, but I would be interested to know.

The use of the images was fascinating and I can see how it may have lead to richer data by breaking down the typical interview process - particularly for those with who likely have strong associations of the norms associated with the format. I think this is a particular strength of the study.

I think the results could be strengthen by restating with the research question/aim to situation the themes. I was a little confused about how the two different sets of themes were related, though this was somewhat cleared up later in the paper.

In terms of discussion, and partly related to the issue outlined with the introduction, but I struggled to understand what is novel and new about the findings. Experience as a variable affecting decision making is widely researched and understood. The authors also themselves state that their findings mirror some of the common models on decision making. I think more needs to be done to explicitly highlight what this paper adds in terms of a) practical implications (which is somewhat done in the conclusion) and b) theoretical implications.

I would also recommend that the restating of the method is removed from the conclusions section and that this section is strengthened so that the contribution is more specific and explicit and less vague.

While I enjoyed and was thankful for the self-reflexivity section, I do think the author (as an insider researcher) needs to say more on how they bracketed their own preconceptions during the data collection and analysis and/or if, as it seems, they leant into their own background, I would appreciate a deeper analysis of the impact this may have had on findings.

A final note, a read through the reference list shows a distinct lack of referencing of most of the decision making models in medicine. Dual process theory is briefly mentioned but cognitive continuum theory (as an example), which is arguably relevant is not. I recommend that the authors dialogue with the existing models more directly to position their studies and findings. There has also been considerable work on individual differences in decision making that goes above and beyond the novice/expert and personality perspectives.

I wish the authors the best in publishing this interesting methodological paper.

Reviewer #2: This manuscript reports on findings from an interesting narrative analyses of interview data from obstetricians and gynecologists in Sweden on their perspectives and experiences of decision making during obstetric emergencies. It is an important area of research, with implications not only for quality of healthcare in obstetric emergencies but potential to add to our understanding of professional decision making processes across a range of healthcare settings. Unfortunately, the research is reported too superficially to realise its potential for understanding decision making even in this specific context, nor the implications for improving decision making processes and outcomes.

The authors set a context for their work as being about understanding variations in intervention rates and outcomes, including potentially harmful over-use of medical intervention. However, this premise is not adequately considered in the subsequent analyses and interpretation. What do the findings offer for uncovering possible explanations? How do they fit (or not) with other research on the possible explanations for these variations and/or overtreatment in terms of obstetric/gynecological decision making?

There are very interesting nuances in the data presented that are missing from the analysis and discussion altogether. For example, the authors indicate at various times that they observed an apparent lack of easy self-awareness and professional reflection when engaging with research participants on this topic (e.g., “motivations and rationales seemed to exist unnoticed”). Is critical self-reflection and debriefing part of routine professional practice (and/or training)? Further, there is an embedded lack of consideration of the patient’s right to autonomy in decision making processes in the participants’ responses and the author’s interpretations. The invisibility of the birthing woman and indicators that her autonomy is largely unacknowledged in the narratives and interpretations (‘could sometimes be included in the discussion’ etc. etc.) may be an integral finding here that goes unmentioned. These findings are of direct relevance to understanding why there may be unwarranted variations in practice and are key to advancing quality decision making in obstetric emergencies. A much deeper and more critical analyses is needed of what is, and perhaps just as importantly what is NOT, embedded in the narratives reported here. Indicators of control and agency, in particular, require deeper analyses and critical reflection.

Related to this, there is insufficient acknowledgement of the role, assumptions and beliefs of the authors and how they may have influenced the research process. The first author/interviewer is noted to be an obstetric and gynecology specialist and senior consultant in perinatology, but much more is needed to understand the researcher bias inherent here. Was the interviewer previously known to participants, and how? What was the power dynamic between researchers and participants? What are the authors’ own fundamental philosophical positions on decision making in obstetrics (including other authors, who are reported to have been involved in ongoing discussions about the findings)? Much stronger reflexivity is needed in both reporting these factors up-front and in consciously acknowledging their influence on analyses and interpretations of the data. The sections of the Discussion on methodological considerations only heighten concerns of bias with a lack of insightful critique on the range inherent assumptions of the research processes.

Interesting that at least one quote about midwifery colleagues was so gendered (see lengthy quote starting page 13 with “It’s a special dynamic”). What was the gender of Physician 9? Are there both cross-professional and cross-gendered power dynamics at play here? Social complexity in this context is mentioned as warranting further research, but could be discussed much more within this work as well.

The context of maternity care organisation in Sweden and the specific role of obstetricians and gynecologists within it would be helpful to a broad readership.

More careful, humanised language is recommended to describe the people being cared for in emergency situations (consider replacing ‘parturient’ with ‘birthing woman’ or ’birthing patient’ if a non-gendered description is preferred). Also consider replacing ‘delivery care’ with ‘intrapartum care’ or ‘care during labour and birth’.

Overall, the implications of these findings could be discussed with much more useful degree of specificity, with some reference to key theories or other research that would be useful in their realisation/implementation.

There are also some typographical/grammatical errors that should be addressed in any subsequent revisions.

6. PLOS authors have the option to publish the peer review history of their article (what does this mean?). If published, this will include your full peer review and any attached files.

Reviewer #1: No

Reviewer #2: **Yes: **Associate Professor Yvette D Miller

---

## [Author Response · Author response to Decision Letter 0]

15 Sep 2021

Reviewer #1: 

General response to reviewer #1’s comments:

We welcome reviewer #1’s valuable and thoughtful comments. The reviewer acknowledges the method(s) and the manuscript for being interesting and conducted with rigor. However, the reviewer underlines that the manuscript needs more work before publication. We have attempted to remedy the concerns in the revised version of the manuscript, especially with regards to contextualizing and justifying the study. We have also expanded on the section regarding self-reflexivity. 

Comments and responses

Thank you for the opportunity to review the paper "decision-making during obstetric emergencies: A narrative approach". Overall, I thought the paper was interesting and the methodology in particular was interesting and conducted with rigor. I do however feel that further work is required before the manuscript is published.

In particular, I feel that more needs to be done to justify why the study was needed. Why is there a need to understand how physicians make sense of and give meaning to their decision-making? And how will understanding their sense making and meaning making processes help us to address a problem/issue. The problem/issue with our current understanding of decision making in emergencies was alluded to but not well defined.

- The introduction has been expanded and the problem/issue somewhat redefined. Because the changes were many, we kindly refer to the manuscript (Line 51 to 92), instead of reproducing the ‘introduction’ text in full.

- A central assumption in the paper was that the disclosure of lived experience through the telling and re-telling of stories would highlight differences in practices (i.e., how physicians make sense out of the complexity of their experience). The narrative approach we used bridges a gap between the individual, understood as agent, and the complexity of human interactions.

It seems like a gap that was trying to be filled was considering team-based decision making (which has evidently been used in anaesthesiology), but if that was the case, why were only Physician’s included? 

- Only physicians (i.e., obstetricians and gynecologists) were included in the study because they:

• “[…] hold a central role in the obstetric team as they are assumed to make key decisions before and during childbirth, especially during emergency situations”. (Line 86 – 88)

Given the plethora of literature on medical decision-making, I felt that the introduction was too short and lacking critical analysis of the existing evidence base and thus lacking the justification for the current study.

- There is indeed a rich literature on medical decision-making. However, it is oftentimes from a nursing context or a non-emergency context or a non-obstetric context or a non-naturalistic context. With the authors having backgrounds in obstetrics and gynecology, narrative science, social science, complexity and systems theory as well as safety science, references were primarily drawn from those domains. In our initial literature review, the causal explanation for the shift between safety and accident, error and correct action, intervention or no-intervention is usually found at one of two opposite poles of a spectrum between individual(s) and system(s). Even groups/teams are either considered as individuals interacting or as a complex system of its own. As referenced in the original article, only a few papers actually manage to keep a complexity perspective on groups/teams in obstetrics without reverting back to a reductionist view. Generally speaking, when considering the individuals, either in isolation or in groups, it is the de-contextualized cognitive (and affective) aspects of their actions/decisions that are in focus. 

Typically, qualitative research is used to answer a research question. I wasn’t clear what the research question was that was being answered in the study.

- The research question was expressed affirmatively as the study’s purpose/aim. The terms ‘exploration’ and ‘how’ refer to a qualitative approach. The bits of sentences ‘make sense of’ and ‘give meaning to’ refer to a narrative approach.

I thought the method and description of the approach was good, though I wasn't clear why median years experience was chosen rather than the traditional mean and standard deviation, which gives more indication of spread, was chosen. Age is also commonly reported though is perhaps less relevant for some reason in this study.

- The median was chosen in the original paper as there were two “outliers” among the participants, one was a trainee with only 3 years of experience and the other was a senior with 31 years of experience. We also intuitively thought that reporting age was less relevant. Nevertheless, age as well as the mean and standard deviation for both age and experience have replaced the median in the revised manuscript: 

• “Mean age: 45 years ± 8.7 years, mean years of experience: 15 ± 9.4 years”. (Line 144 – 145)

Given the study was at 3 different units, I was interested in whether the unit they worked in influenced the themes and style used. Perhaps this was considered and was not of interest as experience drove most of the effect, but I would be interested to know.

- It was mistakenly noted in the original manuscript that participants were from 3 different maternity units. They were in fact from 4 different units. We have corrected this mistake (Line 137). We have also added that:

• “The maternity units used similar guidelines, routines and practices”. (Line 138)

- It is possible that there exist different work cultures influencing the themes and styles. However, this particular question wasn’t the focus of our inquiry. It would be a stretch to speculate on the matter based on the available data. Another complicating factor was the fact that some of the participants had worked at several of the units over the years.

The use of the images was fascinating, and I can see how it may have led to richer data by breaking down the typical interview process - particularly for those with who likely have strong associations of the norms associated with the format. I think this is a particular strength of the study.

- Thank you! We are glad that the method of inquiry triggered your interest.

- Indeed, using art images opened up for a common exploratory space between the interviewer and participants, away from preconceptions about working practices and interview situations.

I think the results could be strengthened by restating the research question/aim to situate the themes. I was a little confused about how the two different sets of themes were related, though this was somewhat cleared up later in the paper.

- The research aim has been restated at the beginning of the result and analysis section:

• “The purpose of this study was to explore how physicians make sense of and give meaning to their decision-making during obstetric emergencies”. (Line 203 – 204) 

- A sentence has also been added to highlight the relationship between themes and decision-making perspectives at an earlier stage:

• “The decision-making perspectives were the expression of physicians’ identities transformed into practice, influenced by the different themes”. (Line 210 – 211)

In terms of discussion, and partly related to the issue outlined with the introduction, but I struggled to understand what is novel and new about the findings. Experience as a variable affecting decision-making is widely researched and understood. The authors also themselves state that their findings mirror some of the common models on decision-making.

- We believe that the novelty of our findings is essentially three-fold: (1) previously observed findings were disclosed through a radically new approach (at least for such a context as the one described in this study), (2) the physicians were both situated within-, and creators of, conditions – for decision-making during obstetric emergencies – in an ongoing dynamically adaptive process, and (3) that this adaptive process was given meaning through an identity expressed as a decision-making perspective. We kindly refer to the revised manuscript (Line 740 to 763), instead of reproducing the ‘conclusions’ section in full.

I think more needs to be done to explicitly highlight what this paper adds in terms of a) practical implications (which is somewhat done in the conclusion) and b) theoretical implications. 

- The practical implications of the findings are mainly twofold. They are important for: (1) the training of individuals and teams, (2) patient-safety and the improvement of care. Indeed, the insights into the physicians’ thinking can be used to better understand why certain decisions were made, possibly guide/mentor in doing things differently, and eventually reorganize work:

• “Overall, the findings have relevance for continuous efforts to improve clinical practice and contribute to our understanding of the variability of strategies and decision-making processes in intrapartum care”. (Line 756 – 758)

- Regarding the theoretical implications, some of the findings confirm previous research:

• “Physicians’ narratives confirm previous findings of multilayered complexity of decision-making in intrapartum care. Various aspects of the reconstructed perspectives are reminiscent of previous findings on individual and team decision-making in emergency medical settings”. (Line 750 – 753)

- There are also implicit, method related findings. Those were mentioned in the original version of the manuscript and have been un-altered in the new version.

I would also recommend that the restating of the method is removed from the conclusions section and that this section is strengthened so that the contribution is more specific and explicit and less vague.

- The restating of the method has been removed from the ‘conclusions’ in the revised version of the manuscript.

- The description of the contribution has been made more explicit. We kindly refer to the previous response on the implication of the findings. 

While I enjoyed and was thankful for the self-reflexivity section, I do think the author (as an insider researcher) needs to say more on how they bracketed their own preconceptions during the data collection and analysis and/or if, as it seems, they leant into their own background, I would appreciate a deeper analysis of the impact this may have had on findings.

- The self-reflexivity section has been expanded in the revised manuscript in order to acknowledge for the interviewer/first author’s (GMR) preconceptions and how they might have impacted data collection and analysis:

• “The fact that he first author (GMR) being was an experienced obstetrician provided certain advantages for the interviews. Acknowledging each other’s expertise within the domain of interest established an atmosphere of trust during the interview situation. Shared contextual understanding was reinforced by the first author further acknowledging participants’ responses, often from having had similar experiences to the ones described and having reflected upon them over the years. This contributed to an ease and flow during the interviews. Participants were enabled to open up more”. (Line 683 – 689), and

• “Being an experienced obstetrician and being inherently embedded within the research context also conferred some difficulties both during the interview and analysis. Indeed, one will always be unable to fully appreciate one’s own assumptions within the domain of interest. In this regard, our understanding and use of both the insider researcher approach and the narrative approach in this paper is grounded in the views of hermeneutic phenomenology [1, 2]. Accordingly, our preconceptions (i.e., knowledge, insights, and experience) of the world come from being inextricably involved within it and “stepping outside” or leaving our biases aside is understood as impossible [1, 3, 4]. However, through a dialogical process of interpretation, preconceptions and the creation of new meanings and an understanding of the world can simultaneously be disclosed [4]”. (Line 692 – 701)

A final note, a read through the reference list shows a distinct lack of referencing of most of the decision-making models in medicine. Dual process theory is briefly mentioned but cognitive continuum theory (as an example), which is arguably relevant is not. I recommend that the authors dialogue with the existing models more directly to position their studies and findings. There has also been considerable work on individual differences in decision making that goes above and beyond the novice/expert and personality perspectives.

- Kenneth Hammond’s cognitive continuum theory has now been mentioned:

• “Another theory worth mentioning here is Hammond’s cognitive continuum theory that models clinical decision making on a spectrum between intuitive and analytic modes [5-8]. Location on the spectrum is dependent on various factors such as type of task at hand (action versus planning), time available (short amount versus greater amount), type of knowledge available (unstructured versus structured), and the kind of health problem needing to be dealt with (acute/unstable versus long-term/stable). The cognitive continuum theory parallels some of the physicians’ narratives about their capacity to modulate their responses during emergencies (i.e., the more urgent the demand for an action was the more intuitive their decision would be)”. (Line 647 – 655)

- However, we have failed to find literature on more decision-making models applicable in a medical emergency setting – including ones taking individual differences into account –other than the ones already mentioned, or else alluded to in the references. Some changes have been made to the manuscript:

• “As a whole, the findings resemble three of four decision-making styles described in a 1995 study by Scott and Bruce [9]. Four decision-making styles were identified through the literature and further tested into an assessment tool. The four styles were: (a) a rational decision-making style characterized by a thorough search for and logical evaluation of alternatives, (b) an intuitive decision-making style characterized by reliance and hunches and feelings, (c) a dependent decision-making style characterized by a search for advice and direction from others and (d) an avoidant decision-making style characterized by attempts to avoid decision making [9]”. (Line 626 – 633), and

• “There is a considerable amount of research on the subject of individual differences (e.g., risk behavior tendency, sensitivity to situational factors, personality, etc.) in relation to decision-making [10]. However, this work has generally been confined to psychology and more specifically to the research field of judgment and decision-making. With a focus on patient safety and optimizing care, the interest in decision-making applied to healthcare has rather been on trying to minimize the impact of individual differences and, to some extent, understand the impact of situational factors”. (Line 668 – 674)

I wish the authors the best in publishing this interesting methodological paper.

- We thank the reviewer for his/her time, comments and encouragement!

Reviewer #2: 

General response to reviewer #2’s comments:

We welcome reviewer #2’s valuable and thoughtful comments. The reviewer acknowledges the method(s) and data for being interesting and that the area of research is important, with the findings having potentially significant implications. However, the reviewer criticizes the analysis and discussion for being too superficial. Eventually, the paper does not help to understand decision-making (in obstetric emergencies), or how it could help improve decision-making processes and outcomes. We have attempted to remedy the concerns in the revised version of the manuscript, especially with regards to the disclosure of preconceptions as well as the philosophical underpinnings of the method(s) and our research approach. 

Comments and responses

This manuscript reports on findings from an interesting narrative analyses of interview data from obstetricians and gynecologists in Sweden on their perspectives and experiences of decision making during obstetric emergencies. It is an important area of research, with implications not only for quality of healthcare in obstetric emergencies but potential to add to our understanding of professional decision making processes across a range of healthcare settings. Unfortunately, the research is reported too superficially to realise its potential for understanding decision making even in this specific context, nor the implications for improving decision making processes and outcomes.

The authors set a context for their work as being about understanding variations in intervention rates and outcomes, including potentially harmful over-use of medical intervention. However, this premise is not adequately considered in the subsequent analyses and interpretation. What do the findings offer for uncovering possible explanations? How do they fit (or not) with other research on the possible explanations for these variations and/or overtreatment in terms of obstetric/gynecological decision making?

- The context has been expanded upon in the revised manuscript. The significance of the findings has been highlighted as well. We kindly refer to the manuscript for the ‘introduction’ (Line 51 to 92) and the ‘conclusions’ (Line 740 to 763) sections.

- Please see the comments to reviewer #1 for more details concerning implications of the findings. 

There are very interesting nuances in the data presented that are missing from the analysis and discussion altogether. 

- The study was essentially descriptive. A thematic narrative analysis outlined by Riessman was used. Narrative analysis interpreted by Czarniawska was also used especially regarding the co-creative and pragmatist approaches of the study. A critical analysis of attitudinal differences in relationship to variations and overtreatment in particular was not the study’s focus. Our intent with the narrative method was to render disparate stories into a coherent whole. This was expanded upon in the ‘study design’ section and the ‘truth and validity’ section of the revised manuscript:

• “Using a thematic narrative analysis outlined by Riessman, and further influenced by Czarniawska’s interpretation, the intent was to eventually render disparate stories into a coherent whole through a dialogical process [11-13]”. (Line 103 – 105), and

• “Here, we subscribe to a pragmatist view [14, 15]. We believe the study has value for both clinical practitioners and researchers alike. Whether this research ultimately contributed anything of value is for the readers to judge [12, 13]”. (Line 726 – 728), and 

• “We believe that both themes and perspectives form coherent wholes that make sense for practitioners”. (Line 730 – 731) 

For example, the authors indicate at various times that they observed an apparent lack of easy self-awareness and professional reflection when engaging with research participants on this topic (e.g., “motivations and rationales seemed to exist unnoticed”). Is critical self-reflection and debriefing part of routine professional practice (and/or training)?

- Swedish obstetricians and gynecologists are invited and encouraged to self-reflect both as trainees and proficient physicians. This is naturally done during clinical rounds and instances of tutoring or when work does not go as planned. Many of the participants’ answers witness of self-awareness and professional reflection to various degrees. However, our interpretation was that the unification of disparate stories into a professional identity driving the physicians’ actions and practice was rather unconscious. Herein lies a paradox.

- A possible explanation for this paradox could be that, after the training period is over and physicians have integrated practices into a way/style that works for them they often work alone (i.e. not with another physician). Peers no longer mirror each other and self-reflection becomes more difficult. 

Further, there is an embedded lack of consideration of the patient’s right to autonomy in decision-making processes in the participants’ responses and the author’s interpretations.

- Although we subscribe to the reviewer’s concern regarding “consideration of the patient’s right to autonomy” in general, we believe this to be less relevant in this particular study, for two reasons:

o (1) It was conducted in a Swedish healthcare setting. Sweden is a country in which trust in governmental institutions, including the healthcare system has traditionally been high. “The patient’s right to autonomy” has been given less outspoken focus in the collective narrative. It is integrated in medical practice through law. The Swedish law on patients’ rights stipulates that: (a) the patient should be treated according to the best available care according to science and clinical experience, (b) the patient has the right to healthcare but do not have the right to choose the care she wants (unless two treatments are equal), (c) the patient can refuse treatment or intervention (except for particular cases when she is suffering of a psychiatric condition, but always after the assessment from a psychiatrist), and (d) treatment should be provided in consultation with the patient and with her consent.

o (2) It was about the particular context of obstetric emergencies. Even if there are variations in the types of emergencies physicians (and midwives) are confronted with, they all happen under a certain time constraint, as was also reported in the participants’ narratives. There is also an assumption of care built into an obstetric emergency, namely that the woman giving birth wants to be a mother (i.e., she want to stay in health to care for her healthy newborn child) and that both physicians and midwives will act in the patient’s best interest. There is certainly a paternalistic bias in this assumption. However, it does not mean that it is wrong. We develop this further in the comment on “the invisibility of the birthing woman”.

o As a side note, we would like to add that malpractice cases are for the great majority handled by a governmental regulatory board, not the courts. Litigations are a rare exception.

The invisibility of the birthing woman and indicators that her autonomy is largely unacknowledged in the narratives and interpretations (‘could sometimes be included in the discussion’ etc. etc.) may be an integral finding here that goes unmentioned.

- An assumed goal driving obstetric care is “a healthy child and mother”. In that sense neither of them are invisible. If by “the invisibility of the birthing woman” the reviewer meant that the woman giving birth was referred to as an object, then the authors agree with the reviewer. This can in fact be the case and was also alluded to in the paper. From a developmental perspective it is only natural that when a practitioner feels less stressed during an emergency, he or she would have more room to empathically connect with the woman/couple, thus considering her/them more as subjects. However, there are at least two problems with this oftentimes-pragmatic approach to obstetric care and particularly to intrapartum care. The first is when the capacity for empathetic connection, or lack thereof is tied to the person (physician or midwife) and/or unexamined practices. The second, related to the first, is when the horizon of care (or even caring) does not reach further than the delivery suite; when the whole life of the woman/couple/family isn’t taken into consideration (e.g., including potential traumatic memories).

- We acknowledge here the lack of feminist critique in the manuscript. However, this wasn’t the focus of our paper. We welcome others to use our rich and original transcriptions in doing so.

These findings are of direct relevance to understanding why there may be unwarranted variations in practice and are key to advancing quality decision making in obstetric emergencies. A much deeper and more critical analyses is needed of what is, and perhaps just as importantly what is NOT, embedded in the narratives reported here. Indicators of control and agency, in particular, require deeper analyses and critical reflection.

- There are indeed two significant weaknesses in the research approach outlined in the study: (1) it is descriptive rather than critical, and (2) it fails to connect narratives and obstetrical outcomes. 

Related to this, there is insufficient acknowledgement of the role, assumptions and beliefs of the authors and how they may have influenced the research process. The first author/interviewer is noted to be an obstetric and gynecology specialist and senior consultant in perinatology, but much more is needed to understand the researcher bias inherent here. 

- The section on ‘self-reflexivity’ has been expanded upon in the revised manuscript. We kindly refer to a previous response to reviewer #1.

Was the interviewer previously known to participants, and how? What was the power dynamic between researchers and participants? What are the authors’ own fundamental philosophical positions on decision making in obstetrics (including other authors, who are reported to have been involved in ongoing discussions about the findings)? Much stronger reflexivity is needed in both reporting these factors up-front and in consciously acknowledging their influence on analyses and interpretations of the data. 

- The relationship of the interviewer to the participants has been added to the manuscript:

• “The interviewer and participants were known to each other on a professional level after having worked together to various degrees over the years”. (Line 147 – 149)

The sections of the Discussion on methodological considerations only heighten concerns of bias with a lack of insightful critique on the range inherent assumptions of the research processes.

- We kindly refer to our response on self-reflexivity to reviewer #1 concerning bias and assumptions. The research process has also been brought up in an earlier response to reviewer #2. In summary, the approach outlined in this paper follows a descriptive, co-creative tradition within narrative analysis. Biases drive an interpretive dialogical process during which the line between the researcher’s and the participants’ ideas tend to get blurred. The goal being the reconstruction of new coherent and sensible narrative. The value of this new narrative is deemed to be ascribed by its readers. This approach differs from other forms of narrative analysis [16] such as ‘Critical Discourse Analysis’ (CDA) and the tradition of Jürgen Habermas for example, which is concerned with analyzing structural relationships of dominance, discrimination, power and control [17]. In contrast to the approach in this study, in CDA the researcher needs to consider the data from at a distance.

Interesting that at least one quote about midwifery colleagues was so gendered (see lengthy quote starting page 13 with “It’s a special dynamic”). What was the gender of Physician 9? 

- Physician 9 is a middle-aged woman with substantial experience in obstetrics. She was particularly reflective on her experience and sensitive to the use of pictures.

- Gendering the quotes does not provide more nuances, more depth or more coherence to the narrative as a whole in this study.

Are there both cross-professional and cross-gendered power dynamics at play here?

- A critical assessment of both cross-professional and cross-gendered power dynamics was not the focus of this study. However, the new section on childbirth in Sweden includes the following section:

• “Historically, a non-interventionist ideal, i.e., wait and see rather than intervene, has guided practice for both midwives and obstetricians in Sweden [18]. The relationship between physicians and midwives in Sweden has also been characterized by considerably more teamwork rather than conflicts [18]”. Line (131 – 134)

- However, cross-professional power dynamics have been alluded to in previous research as well [19] and cross-gender power dynamics most certainly exist within the world of Swedish obstetrics. Some these aspects embedded in practice are formally discussed in the midwifery education or during physicians’ training in obstetrics. 

Social complexity in this context is mentioned as warranting further research, but could be discussed much more within this work as well.

- The sentence: “Further research on social complexity in this domain is also warranted” (Line 612) has been removed.

The context of maternity care organisation in Sweden and the specific role of obstetricians and gynecologists within it would be helpful to a broad readership.

- A specific section about maternity care organization in Sweden has been added to the revised version of the manuscript:

• “Intrapartum care in Sweden is institutionalized and publicly funded. There are 40 maternity units, and about 115.000 to 120.000 births per year serving a population of 10,2 million (2019) [20]. Pregnancy, labor, and childbirth are first and foremost considered natural processes. In 2020, 88% of births were so called normal births. Normal births are births without greater intervention: no Caesarean sections or no VE/forceps, postpartum bleeding below 1500 mL or no need for blood transfusion, no sphincter rupture, no Apgar score below 7 at 5 minutes [20]. The care is standardized by national and local guidelines. Trained and autonomous midwives follow most of patients during pregnancy and assist during labor and childbirth. Midwifery is a university degree constructed around the medical sciences and modern medicine, and a previous degree in nursing is required. Physicians, trainees or specialists in obstetrics and gynecology are primarily involved in the care of patients with complicated pregnancies or when complications occur during labor and childbirth. Physicians take over the medical responsibility from the midwife when getting involved during pregnancy, labor, or childbirth.” (Line 119 – 131)

More careful, humanised language is recommended to describe the people being cared for in emergency situations (consider replacing ‘parturient’ with ‘birthing woman’ or ’birthing patient’ if a non-gendered description is preferred). Also consider replacing ‘delivery care’ with ‘intrapartum care’ or ‘care during labour and birth’.

- The necessary changes have been made to the revised manuscript.

Overall, the implications of these findings could be discussed with much more useful degree of specificity, with some reference to key theories or other research that would be useful in their realisation/implementation.

- We kindly refer to one of the previous responses to reviewer #1 concerning the implications of the findings.

There are also some typographical/grammatical errors that should be addressed in any subsequent revisions.

- The necessary changes have been made to the revised manuscript.

- We thank the reviewer for her time, comments and encouragement!

References

1. Laverty SM. Hermeneutic phenomenology and phenomenology: A Comparison of historical and methodological considerations. Int J Qual Methods. 2003;2(3):21-35.

2. Salvador JT. Revisiting the philosophical underpinnings of qualitative research. Int educ res. 2016;2(6):4-6.

3. Tufford L, Newman P. Bracketing in Qualitative Research. Qual Soc Work. 2010;11(1):80-96.

4. Alvesson M, Sandberg J. Pre-understanding: An interpretation-enhancer and horizon-expander in research. Organ Stud. 2021;10.1177/0170840621994507.

5. Hamm RM. Clinical intuition and clinical analysis: Expertise and the cognitive continuum. In: Dowie JA, Elstein AS, editors. Professional judgment: A reader in clinical decision making. New York, NY: Cambridge University Press; 1988. p. 78-105.

6. Standing M. Clinical judgement and decision-making in nursing - nine modes of practice in a revised cognitive continuum. J Adv Nurs. 2008;62(1):124-34.

7. Standing M. Clinical judgement and decision-making in Nursing and interprofessional healthcare. 1st ed. Berkshire (UK): Open University Press, McGraw-Hill; 2010.

8. Parker-Tomlin M, Boschen M, Morrissey S, Glendon I. Cognitive continuum theory in interprofessional healthcare: A critical analysis. J Interprof Care. 2017;31(4):446-54.

9. Scott SG, Bruce RA. Decision-Making Style: The Development and Assessment of a New Measure. Educ Psychol Meas. 1995;55(5):818-31.

10. Appelt KC, Milch KF, Handgraaf MJJ, Weber EU. The Decision Making Individual Differences Inventory and guidelines for the study of individual differences in judgment and decision-making research. Judgm Decis Mak. 2011;6(3):252-62.

11. Riessman CK. Narrative analysis. In: Lewis-Beck MSB, Alan; Futing Liao, Tim editor. The Sage Encyclopedia of Social Science Research Methods. 3. Thousand Oaks, CA: SAGE Publications, Inc.; 2003.

12. Riessman CK. Narrative methods for the human sciences. Thousand Oaks (CA): SAGE Publications, Inc; 2008.

13. Czarniawska B. Narratives in social science research. London (UK): SAGE Publications Ltd; 2004.

14. Rorty R. The pragmatist’s progress. In: Collini S, editor. Interpretation and overinterpretation - Umberto Eco. Cambridge (UK): Cambridge University Press; 1992. p. 89-108.

15. Topper K. In defence of disunity: Pragmatism, hermenustics and the social sciences. Polit Theory. 2000;28(4):509-39.

16. Landman T. Phronesis and narrative analysis. In: Flyvbjerg B, Landman T, Schram S, editors. Real social science: Applied phronesis. Cambridge, UK: Cambridge university press; 2012. p. 308.

17. Wodak R, Meyer M. Methods of critical discourse analysis. 4th ed. London (UK): SAGE Publications, Inc; 2005.

18. Milton L. Midwives in the Folkhem: Professionalisation of Swedish midwifery during the interwar and postwar period [Dissertation]. Uppsala: Uppsala University; 2001.

19. Bergström J, Dekker S, Nyce JM, Amer-Wåhlin I. The social process of escalation: a promising focus for crisis management research. BMC Health Serv Res. 2012;12:161.

20. Graviditetsregistret 2021 [The Swedish Pregnancy Register is a Certified National Quality Register initiated by the Swedish Healthcare. It collects and processes information all the way from early pregnancy to a few months after birth]. Available from: https://www.medscinet.com/gr/default.aspx.

---

## [Decision Letter · Decision Letter 1]

8 Nov 2021

Decision-making during obstetric emergencies: A narrative approach

PONE-D-21-12199R1

Dear Dr. Raoust,

We’re pleased to inform you that your manuscript has been judged scientifically suitable for publication and will be formally accepted for publication once it meets all outstanding technical requirements.

Kind regards,

Bernadette Watson, Ph.D.

Academic Editor

PLOS ONE

Additional Editor Comments (optional):

You have worked hard to address Reviewer 1 and had already answered Reviewer 2 in the first revision.

Reviewers' comments:

Reviewer's Responses to Questions

**Comments to the Author**

1. If the authors have adequately addressed your comments raised in a previous round of review and you feel that this manuscript is now acceptable for publication, you may indicate that here to bypass the “Comments to the Author” section, enter your conflict of interest statement in the “Confidential to Editor” section, and submit your "Accept" recommendation.

Reviewer #1: All comments have been addressed

2. Is the manuscript technically sound, and do the data support the conclusions?

Reviewer #1: Yes

3. Has the statistical analysis been performed appropriately and rigorously? 

Reviewer #1: N/A

4. Have the authors made all data underlying the findings in their manuscript fully available?

Reviewer #1: Yes

5. Is the manuscript presented in an intelligible fashion and written in standard English?

Reviewer #1: Yes

6. Review Comments to the Author

Reviewer #1: I would like to thank the authors for their full consideration of the points I raised in the initial review. I thought that they had been addressed in a meaningful way and I thoroughly enjoyed reading the revised manuscript.

7. PLOS authors have the option to publish the peer review history of their article (what does this mean?). If published, this will include your full peer review and any attached files.

Reviewer #1: **Yes: **Nicola Sheeran

---

## [Editor Report · Acceptance letter]

17 Jan 2022

PONE-D-21-12199R1 

Decision-making during obstetric emergencies: A narrative approach 

Dear Dr. Raoust:

I'm pleased to inform you that your manuscript has been deemed suitable for publication in PLOS ONE. Congratulations! Your manuscript is now with our production department. 

Kind regards, 

on behalf of

Dr. Bernadette Watson 

Academic Editor

PLOS ONE